# Preparation and Characterization of Gold Nanorods Coated with Gellan Gum and Lipoic Acid

**Paola Varvarà** [1] , **Luigi Tranchina** [2], **Gennara Cavallaro** [1] **and Mariano Licciardi** [1,*]

1   Department of Scienze e Tecnologie Biologiche Chimiche e Farmaceutiche (STEBICEF),
    Università degli Studi di Palermo, 90123 Palermo, Italy; paola.varvara@unipa.it (P.V.);
    gennara.cavallaro@unipa.it (G.C.)
2   Advanced Technologies Network Center (ATeN Center), Università degli Studi di Palermo, 90128 Palermo,
    Italy; luigi.tranchina@unipa.it
*   Correspondence: mariano.licciardi@unipa.it; Tel.: +39-091-23891927

**Abstract:** Gold nanorods (AuNRs) can combine therapeutic hyperthermia with diagnostic features, representing a smart choice to address personalized cancer treatments. In this regard, a crucial quest is the selection of the right biocompatible coating agent able to stabilize them in the physiological environment, further endowing the possibility to load bioactive molecules and/or targeting moieties. Therefore, AuNRs optical properties can be successfully merged with advantageous materials features to obtain selective photothermal therapy (PTT) systems. Here, the natural materials lipoic acid (LA) and the polysaccharide gellan gum (GG) were chosen to prepare three types of stabilized gold nanorods, using LA (AuNRs/LA), a layered coating of LA and GG (AuNRs/LA,GG) or a newly synthesized covalent derivative of LA and GG (AuNRs/GG-LA). The samples displayed diverse stability and dispersibility. Hydrodynamic diameters and surface potential analyses confirmed the nanometric size (100–200 nm) and showed surface charges ranging from +19.5 to −25.6 mV. Particular attention was thus paid to analyze the differences between hyperthermia properties exhibited after near-infrared (NIR) laser irradiation. Furthermore, the cytocompatibility and photothermal effect were tested on HCT116 human colon cancer cell line. Collected data have finally allowed selecting AuNRs/LA,GG as the best candidate for possible use in PTT of cancer.

**Keywords:** gold nanorods; polysaccharide coating; gellan gum; lipoic acid; hyperthermia; photothermal therapy

## 1. Introduction

Nanomaterials science constantly provides new approaches to overcome the limitations that lie in disparate fields of current technology. Biomedical research represents among the top areas that take advantage of nanomaterials' progress. As a consequence, there has been recently registered an expanding production of new nanometric devices that cover a wide panorama of biomedical applications such as targeted drug delivery, biosensing, and bioimaging [1–3]. In this scenario, a clear breakthrough has been achieved exploiting inorganic nanostructures, which endow to synergistically combine the advantageous dimensional characteristics of nanomaterials with their unique features including hyperthermia, conductivity, magnetism, and contrast properties [4,5]. For these reasons, diverse nanometric inorganic materials, such as superparamagnetic iron oxide nanoparticles (SPIONs) [6,7], graphene oxide [8,9], silica nanoparticles [10,11], quantum dots [12,13], colloidal gold [14,15], have been already successfully employed for drug delivery or theranostic purposes.

Gold nanoparticles (AuNPs) are highly versatile inorganic nanoparticles that have paved the way for a wide range of new biomedical applications, with a special focus on cancer treatment.

By modulating the synthesis procedures and parameters, AuNPs can be easily produced in a large variety of shapes and sizes, displaying different properties and usage [16–18]. As a noble metal, gold in nanometric size exhibits localized surface plasmon resonance (LSPR), a peculiar feature responsible for its optical behavior. When gold nanoparticles are irradiated with electromagnetic radiation with the wavelength corresponding to the LSPR band, free electrons oscillate harmoniously and in a localized manner around the particle lattice. Because of this phenomenon, diffuse electromagnetic waves can be used for the acquisition of diagnostic images using imaging techniques such as computed tomography (CT) or confocal microscopy. In addition, the energy of the absorbed electromagnetic radiation can be converted into heat, and for this reason, gold colloids are useful agents in photothermal therapy (PTT) of cancer. LSPR bands of AuNPs lay in the UV-Visible-Near InfraRed region (UV–VIS–NIR) although their definite position and number are dependent on the size, shape, and state of aggregation of the nanoparticles. It follows that the modification of the synthesis parameters allows to obtain AuNPs with tailored optical properties [19,20]. In this context, cylindrical gold nanoparticles (nanorods-AuNRs) have gained increasing interest due to the facile synthesis and the position of their longitudinal LSPR located in the NIR region (wavelength: 650–900 nm), which overlaps the biological window where the absorption of water and hemoglobin (and hence of body tissues) is minimized. Therefore, the use of AuNRs allows to combine the possibility to obtain a targeted delivery (via Enhanced Permeability and Retention effect and/or active targeting), as well as the ability to exert cancer PTT with superior penetration and efficacy.

Although colloidal gold stands out for its bioinertia, the use of capping agents is fundamental for biomedical applications [21]. The coating operation is manifold because it can stabilize AuNRs creating a protective hydrophilic film that prevents aggregation in the aqueous physiological environment, further replacing cytotoxic stabilizing agents used during the synthesis procedure. Besides, the choice of appropriate coating agents allows to increase cytocompatibility, as well as to load therapeutic molecules and active targeting functionalities, obtaining a targeted drug delivery.

Lipoic acid (LA) is a small molecule naturally found in the human body that possesses a metal chelating activity and proven ROS scavenging ability [22–24]. From the structural point of view, LA is characterized by an intracycle disulfide group capable to interact with the gold surface via gold-sulfur chemistry [25,26] and an ionizable carboxylic terminal, that confer good overall hydrophilicity.

AuNRs-based drug delivery systems may be properly stabilized by small molecules, nevertheless, to increase cytocompatibility and drug loading capacity, a biocompatible polymeric coating is usually adopted. Polymeric hydrophilic or amphiphilic materials have been widely explored as AuNPs capping agents. Among them are not only synthetic polymers such as polyethylene glycol PEG-SH [27] and polyaminoacidic structures [15] but also natural polymers such as proteins [28] and polysaccharides [14]. Gellan gum (GG) is a linear polysaccharide consisting of a tetrameric repeating unit of ((D-Glucose)-(D-Glucuronic Acid)-(D-Glucose)-(L-Rhamnose)), with a negative net charge. It is synthesized by Sphingomonas elodea and it displays water solubility and in vivo biodegradability [29]. GG is a food additive approved by Food and Drug Administration (FDA) [30] and, as a polysaccharide, is characterized by the presence of carbohydrate units rich in hydroxyl groups able to establish hydrogen bonds with different classes of molecules [31].

This work aimed to test promising natural materials as coating agents of gold nanorods, focusing on how the choice of the material, as well as the coating strategy, could influence physicochemical properties and PTT efficacy of the final nanostructure. Concretely, three types of coated gold nanorods were prepared using lipoic acid (LA) and gellan gum (GG) as capping agents. In particular, AuNRs/LA, AuNRs/LA,GG, and AuNRs/GG-LA were produced using, respectively, the small disulfide molecule LA, a layer by layer coating of LA and polysaccharide chains of GG, or the covalent derivative obtained from esterification between GG and LA as stabilizing agents. The prepared AuNRs have been characterized by evaluating their stability in aqueous medium over time and monitoring possible aggregation through UV–VIS spectrometry. The size and surface properties as well as hyperthermia abilities after NIR irradiation were then investigated and finally, biological studies were performed

on human colorectal cancer cell line (HCT116), to assay coated nanorods cytocompatibility and hyperthermia effect in vitro. The results obtained were thus compared to select the most promising candidate for potential use in cancer PTT.

## 2. Materials and Methods

### 2.1. Materials

Tetrachloroauric(III) acid trihydrate (HAuCl$_4$ 3H$_2$O), silver nitrate (AgNO$_3$ ≥ 99.0%), α-lipoic acid (LA), hexadecyltrimethylammonium bromide (CTAB ≥ 96%), ascorbic acid (≥99%), sodium borohydride (NaBH$_4$ 99%), triethylamine (TEA), *N*-(3-dimethylaminopropyl)-*N'*-ethylcarbodiimide hydrochloride (EDC-HCl), *N*-hydroxysulfosuccinimide sodium salt (NHSS), Dulbecco's phosphate saline buffer (DPBS) and Gold Test for gold determination were purchased from Merck, Milan, Italy. The gellan gum (GG, Mw ≈ 50 KDa) used was obtained starting from GELZAN (Merck, Milan, Italy) through a degradation process in basic catalysis already known in the literature [32]. The Texas Red Hydrazide Invitrogen probe and the LIVE/DEAD mammal cell kit were purchased from Thermo Fisher (Eugene, OR, USA). MTS 3-[4,5-dimethylthiazol-2-yl]-5-(3-carboxymethoxyphenyl)-2-(4-sulphophenyl)-2H-tetrazolium] reagent (CellTiter 96® AQueous One Solution Cell Proliferation,) was purchased from Promega (Madison, WI, USA) and used according to the manufacturer's instructions. Spectra/Por dialyses tubing was purchased from Spectrum Laboratories Inc. (Irving, TX, USA). Sephadex G-15 gel permeation resin was purchased from Fluka (Buchs, Switzerland). Milli-Q water (resistivity 18.2 MΩ cm at 25 °C) was used in all experiments. All reagents and solvents were of analytical grade unless otherwise specified.

Human colon carcinoma cells (HCT116) were purchased from Istituto Zooprofilattico Sperimentale della Lombardia e dell'Emilia Romagna "B. Ubertini" (Brescia, Italy) and cultured in DMEM, containing 10% (*v/v*) of fetal bovine serum (FBS), 100 U of penicillin G, 100 mg·mL$^{-1}$ of streptomycin, and 2 mM of L-glutamine (Euroclone, Milan, Italy), at 37 °C in 5% CO$_2$ humidified atmosphere.

### 2.2. Synthesis and Characterization of the Gellan Gum Derivative GG-LA

The grafting of lipoic acid (LA) onto the gellan gum (GG) backbone was carried out in aqueous environment at controlled pH and temperature. More specifically, 100 mg of gellan gum sodium salt (GG, Mw ≈ 50 KDa) was solubilized in 3 mL of bidistilled water. Separately, lipoic acid (LA to GG repeating units' molar ratio equal to 0.3) was dispersed in ultrapure water and dissolved adjusting the pH to 6.8 with TEA. The activation of LA occurred by addition of EDC-HCl and NHSS (mol EDC-HCl≡NHSS/mol LA equal to 1.2:1) and the pH was once again adjusted to 6.8. The reaction mixture was kept stirring at 40 °C for 18 h, checking the pH for the first 2 h. Unreacted chemicals were removed through dialysis tubing (cut-off 12–14 KDa) and then, the product was further purified by gel permeation chromatography using Sephadex G-15 resin. The final derivative, called GG-LA, was collected after freeze-drying (yield: 85% calculated on the starting GG weight). Here, $^1$H NMR was (300 MHz, D$_2$O/NaOD, 25 °C, δ): 1.2 (3H GG, -CH$_3$ of rhamnose), 1.55 and 2.2 (4H LA, -HO-CO-CH$_2$-CH$_2$-CH$_2$-CH$_2$-cCH-CH$_2$-CH$_2$-S-S-), 2.23 (2H LA -HO-CO-CH$_2$-CH$_2$-CH$_2$-CH$_2$-cCH-CH$_2$-CH$_2$-S-S-), 2.97 (2H LA, -HO-CO-CH$_2$-CH$_2$-CH$_2$-CH$_2$-cCH-CH$_2$-CH$_2$-S-S-), 3.1–4.1 (1H GG, -CH of glucose), 5.03 (1H GG, -CH of glucuronic acid), 5.2 (1H GG, -CH of rhamnose).

SEC analyses were performed using a Phenomenex PolySep-GFC-P4000 (Torrance, CA, USA) column connected to a light scattering detector (LALS, 15° and RALS, 90°) and a refractive index detector (Agilent 1260 Infinity). TRIS buffer 0.15 mM at pH 9 was used as mobile phase and the flow was set at 0.8 mL·min$^{-1}$. Samples were incubated overnight in mobile phase under stirring (15 mg·mL$^{-1}$) and filtrated through a syringe filter (0.45 μm cut-off) before injection.

### 2.3. Synthesis and Scanning Electron Microscopy (SEM) of CTAB-Stabilized AuNRs (CTAB-AuNRs)

To produce the coated systems, CTAB-AuNRs synthesized as previously reported [15] were used as starting material. Briefly, CTAB-AuNRs were produced through seed-mediated growth procedure. In a first step, gold seeds were prepared via $NaBH_4$ (300 µL, 0.01 M) reduction in Au III from $HAuCl_4$ (25 µL, 0.05 M), under vigorous stirring and in the presence of hexadecyltrimethylammonium bromide (CTAB 0.1 M) as stabilizing agent. The seeds were then allowed to elongate, reaching a cylindrical shape, in a growth CTAB solution (0.1 M) containing 500 µL of 0.05 M $HAuCl_4$ and 950 µL of HCl 1 M in which 600 µL $AgNO_3$ 0.01 M, 400 µL of ascorbic acid 0.1 M, and 120 µL of gold seeds were added. After incubation at 28 °C, CTAB-stabilized AuNRs (CTAB-AuNRs) were thus obtained.

Scanning electron microscopy (SEM) was performed to investigate the size and morphology of CTAB-AuNRs and AuNRs/LA,GG, using a scanning electron microscope ESEM Philips XL30. CTAB-AuNRs in aqueous dispersion were washed 2 times with Milli-Q water to remove the excess surfactant, placed on a TEM copper grid (10 µL, 0.001 mg·mL$^{-1}$) and allowed to dry for 24 h. SEM analyses were carried out after deposition of the grid on a double-sided adhesive tape, previously applied on a stainless-steel stub.

### 2.4. Preparation of Coated AuNRs (AuNRs/LA, AuNRs/LA,GG, and AuNRs/GG-LA)

In a typical AuNRs/LA preparation, 20 mL of CTAB-AuNRs were washed 3 times with Milli-Q $H_2O$ and recovered by centrifugation (10,000 rpm, 5 min, and 24 °C). Subsequently, lipoic acid (LA) and glucose were added under stirring to the dispersion of washed AuNRs, following the Au:LA:glucose molar ratios equal to 1:2:2. The dispersion was thus neutralized adjusting the pH to 7 with NaOH. The mixture was kept incubating for 18 h at room temperature, washed one last time with ultrapure water, and the initial volume was restored by adding the same solvent.

AuNRs/LA,GG were produced from AuNRs/LA by incubation with gellan gum (GG). In particular, after the last washing of the AuNRs/LA, the pellet was slowly dripped under stirring into a polymeric dispersion of GG in Milli-Q water (1 mg·mL$^{-1}$; pH 7) and maintained at room temperature under stirring for 18 h. To formulate AuNRs/LA,GG, an Au: GG weight ratio of 1:10 was set.

Finally, for the preparation of AuNRs/GG-LA, 20 mL of CTAB-AuNRs was washed 4 times and recovered by centrifugation (10,000 rpm, 5 min, 24 °C). The pellet was then added dropwise into a dispersion of GG-LA in Milli-Q water (1 mg·mL$^{-1}$) and glucose (mol/mol ratio Au/glucose equal to 1:2), neutralized with NaOH, and incubated 18 h at room temperature under stirring. The coating with GG-LA was accomplished setting an Au:GG-LA weight ratio of 1:10.

The Au concentration of CTAB-AuNRs was calculated by UV–VIS analysis by measuring the absorbance of the sample at the fixed wavelength of 400 nm [33].

### 2.5. UV–VIS–NIR Spectroscopy, Stability Studies, and Reconstitution Tests

AuNRs/LA, AuNRsLA,GG, and AuNRs/GG-LA were analyzed using Shimadzu UV–VIS–NIR 2400 spectrometer. The sample spectra were acquired in water dispersion (0.5 mg·mL$^{-1}$) at room temperature investigating the wavelength range between 200 and 900 nm. Employing the same conditions, the absorbance spectra of the samples were recorded for 30 days at 1-week intervals, in order to evaluate their stability in aqueous dispersion as a function of time.

To perform reconstitution tests, 7 mL of sample (AuNRs/LA, AuNRsLA,GG, or AuNRs/GG-LA) was freeze-dried and dispersed again in an equal volume of Milli-Q water, then characterized through UV–VIS–NIR spectroscopy setting the same parameters described for stability tests.

### 2.6. Dynamic Light Scattering Analyses, Zeta Potential Measurements, and Determination of Gold Content

The entire set of coated systems (AuNRs/LA, AuNRsLA,GG, and AuNRs/GG-LA) was characterized in terms of hydrodynamic diameters, polydispersity index (PDI), and Zeta potentials, using the Malvern Zetasizer NanoZS instrument, equipped with a 532 nm laser with the fixed angle

of 173°. Aqueous dispersions of the samples (Au 0.025 mg·mL$^{-1}$) were prepared and analyzed at 25 °C. The hydrodynamic diameter and the PDI were obtained by examining the cumulants of the correlation function. The Zeta potential (mV) was calculated from electrophoretic mobility using the Smoluchowsky relationship and assuming Ka ≫ 1 (where K and a are, respectively, the Debye–Hückel parameter and the particle radius).

The gold content was determined after complete oxidation of samples aqueous dispersions in an HCl 37%/HNO$_3$ 69.5% 3:1 v/v mixture, using Spectroquant$^®$ Gold Test (Merck, Milan, Italy) and expressed as a concentration (mg·mL$^{-1}$).

## 2.7. Synthesis of LA-TRH Derivative and LA Determination Content in AuNRs/LA-TRH Systems

A fluorescent dye was used to label lipoic acid in order to determine LA content in AuNRs/LA. First of all, LA was dispersed in Milli-Q water (0.49 mg·mL$^{-1}$) and solubilized adjusting pH to 6 with NaOH. Subsequently, EDC·HCl and NHSS were added, setting EDC·HCl/NHSS:LA molar ratios equal to 1.2:1. A solution of Texas Red Hydrazide (TRH) probe in H$_2$O/DMF 8:2 was thus added dropwise to the previously prepared dispersion (mol/mol TRH/LA ratio equal to 1.13:1), the pH was adjusted to 4.5, and the mixture was maintained to react 18 h at room temperature. The labeled product obtained (LA-TRH) was then used to prepare AuNRs/LA-TRH. Specifically, a dispersion of CTAB-AuNRs was washed 3 times with ultrapure water, recovered by centrifugation (10,000 rpm, 5 min, 24 °C), and mixed together with LA-TRH and glucose (mol/mol LA-TRH/glucose equal to 1:1). After neutralization with NaOH, the flask content was kept incubating 18 h at room temperature, purified by centrifugation (10,000 rpm, 5 min, and 24 °C) and, finally, the initial volume was restored using Milli-Q water.

The amount of LA-TRH was determined by evaluating the fluorescence of AuNRs/LA-TRH samples. The labeled system was analyzed using an Eppendorf PlateReader AF2200 spectrophotometer (ex: 530 nm and em: 590 nm) and comparing the fluorescence recorded with a calibration curve obtained from standard solutions of TRH probe (linearity range: $5 \times 10^{-4}$–$1 \times 10^{-5}$ mg·mL$^{-1}$; R$^2$ = 0.998).

## 2.8. X-ray Photoelectron Spectroscopy (XPS) Analyses on AuNRs/LA

To calculate LA content of AuNRs/LA,GG sample, X-ray photoelectron spectroscopy (XPS) was performed. Analyses were conducted on freeze-dried samples using a PHI 5000 VersaProbe II spectrometer (ULVAC-PHI, Inc., Kanagawa, Japan) with an Al Kα source, 1486.6 eV, 200 μm, and 50 W.

## 2.9. Evaluation of Coated Gold Nanorods Hyperthermia

AuNRs/LA, AuNRs/LA,GG, and AuNRs/GG-LA samples at different Au concentrations (10, 20, 35 μg·mL$^{-1}$) were placed in a 24-well plate and treated with a diode laser (GBox 15A/B by GIGA Laser) with the fixed wavelength of 810 nm, setting the laser power of $3.5 \times 10^{-3}$ W/mm$^3$, $7 \times 10^{-3}$ W/mm$^3$, and $14 \times 10^{-3}$ W/mm$^3$. A fiber optic temperature detector (±1 °C sensitivity) was used to record temperatures at preset laser exposure time ranging from 0 to 300 s, and hyperthermia profiles were thus obtained plotting temperatures versus exposure time.

Infrared-camera hyperthermia studies were performed using a FlirT250 Infrared Camera with a resolution of 240 × 180 pixels, a sensitivity of 80 mK NETD/0.08 °C and a measurable temperature range of −20 °C : 50 °C. In particular, dispersions of AuNRs/LA, AuNRs/LA,GG, and AuNRs/GG-LA at gold concentrations of 2.5 and 5 μg·mL$^{-1}$ were placed in 24-well plates and treated for 100 s with the aforementioned diode laser, setting the power of $10 \times 10^{-3}$ W/mm$^3$. At time intervals of 20 s, the IR images were acquired, and the temperatures were recorded. The same experiment was performed on an equal amount of ultrapure water, used as negative control.

## 2.10. Cytocompatibility Tests

Cell viability of human colon cancer cells (HCT116) after incubation with AuNRs/LA, AuNRs/LA,GG, and AuNRs/GG-LA was evaluated using the MTS viability assay (Cell Titer 96 Aqueous One Solution Cell Proliferation assay, Promega, Madison, WI, USA). HCT116 cell line was cultured in

supplemented DMEM, seeded in 96-well plates at the density of $2 \times 10^4$ cells per well and placed for 24 h at 37 °C (humidified environment with 5% $CO_2$), to allow adhesion. The well medium was then replaced with 200 μL nanorods dispersion in DMEM at concentrations equal to 1, 2.5, or 5 μg·mL$^{-1}$, and the plates were placed again at 37 °C. After 24 or 48 h, the well content was removed, and 120 μL MTS solution in DMEM was added in each well (MTS dilution: sixfold). Plates were then incubated for further 2 h at 37 °C and the absorbance of the samples (λ: 492 nm) was read using a microplate reader (Eppendorf PlateReader AF2200). Cell viability was expressed as a percentage of viability compared to untreated cells used as negative control.

### 2.11. In Vitro Evaluation of the Hyperthermal Effect on HCT116 Cell Line

Human colon cancer cells (HCT116) were seeded in 24-well plates at the density of $8 \times 10^4$ cells per well, in supplemented DMEM and maintained at 37 °C (5% $CO_2$) for 24 h. Cells were then incubated with AuNRs/LA, AuNRs/LA,GG, and AuNRs/GG-LA dispersions in DMEM at the concentration of 5 μg·mL$^{-1}$ and placed again at 37 °C for 24 h. Subsequently, cells were treated with a diode laser (λ = 810 nm) setting the power of $10 \times 10^{-3}$ W/mm$^3$ for 20, 40, 60, 80, or 100 s. On completion, the supernatant was discarded and substituted with 450 μL of DMEM and 50 μL of MTS solution. Cells were thus incubated for 2 h at 37 °C and the absorbance at 492 nm was read in each well using a microplate reader (Eppendorf PlateReader AF2200). The same procedure was applied to untreated cells, with or without laser treatment. Cell viability was calculated as a percentage with respect to the untreated cells used as control.

The cytotoxic efficacy of the tested systems was also qualitatively assessed by performing the LIVE/DEAD assay (Thermo Fischer, Eugene, OR, USA) to discriminate living cells from seriously damaged or dead ones. In particular, the experiment was carried out on cell samples ($8 \times 10^4$ cells per well) incubated with AuNRs/LA, AuNRs/LA,GG, and AuNRs/GG-LA and treated for 100 s with a diode laser (λ = 810 nm; power = $10 \times 10^{-3}$ W/mm$^3$). LIVE/DEAD assay was carried out following the manufacturer's specifications. Images were acquired by a fluorescence microscope using an Axio Cam MRm (Zeiss, Oberkochen, Germany).

### 2.12. Statistical Analysis

Statistical analysis was performed using Student's two-tailed *t*-test. Statistical significance was established following the criterion * $p < 0.05$. All values are the average of three experiments ± standard deviation.

## 3. Results and Discussion

### 3.1. Synthesis of GG-LA Derivative

In order to obtain a comparison between the physicochemical properties and the hyperthermia efficacy of gold nanorods stabilized by different coating agents, the ester derivative GG-LA was first synthesized starting from the sodium salt of the natural polysaccharide gellan gum (GG; 50 KDa). The synthesis of the GG-LA derivative was carried out in an aqueous medium at controlled temperature and pH (Scheme 1).

GG complete dispersion in Milli-Q water was achieved after 3 cycles of sonication/stirring in a water bath at 40 °C (10 min/5 min). The dispersion of high-molecular-weight GG normally requires medium-high temperatures (90 °C) [32], but in this case, the low molecular weight of the starting polymer has increased its handling, allowing its dispersion in milder conditions. Lipoic acid (LA) was placed in Milli-Q water and its solubilization was encouraged by adjusting the pH up to 6.8. The choice of the molar ratio between LA and repetitive units of GG (0.3) was made with the aim to guarantee an adequate quantity of disulfide groups available to establish a proper interaction with the gold surface. The activation of the carboxylic groups of LA was carried out via carbodiimide reaction (EDC·HCl/NHSS), and therefore, the formation of the ester bond took place at pH 6.8, keeping the

reaction flask at 40 °C for 18 h. After purification, the GG-LA derivative was recovered by freeze-drying and characterized by $^1$H NMR spectroscopy (Figure 1).

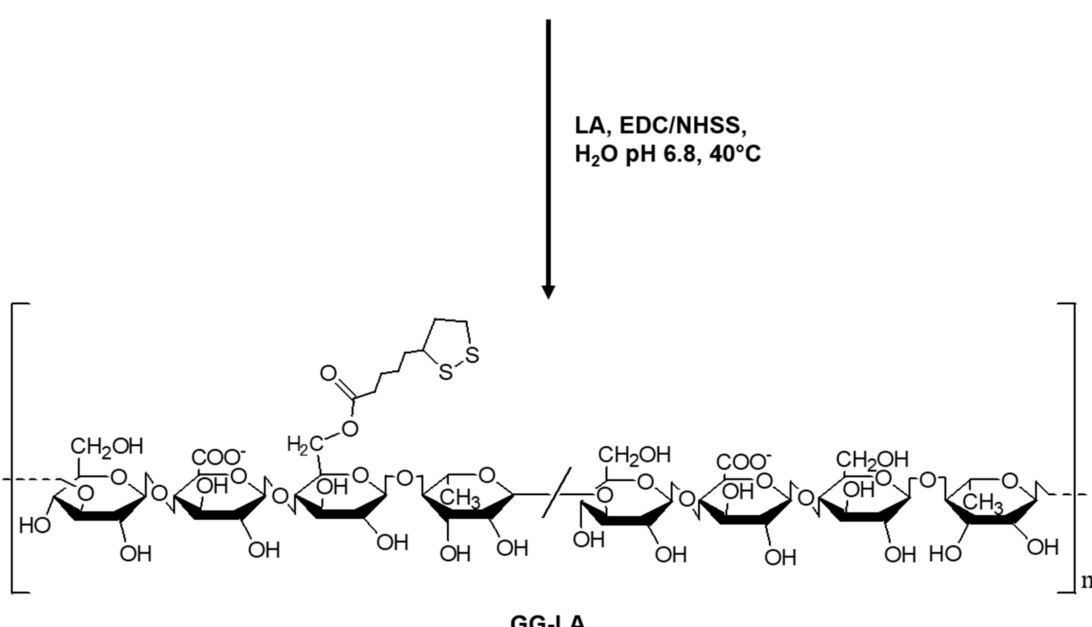

**Scheme 1.** Synthesis of the ester derivative of gellan gum and lipoic acid (GG-LA).

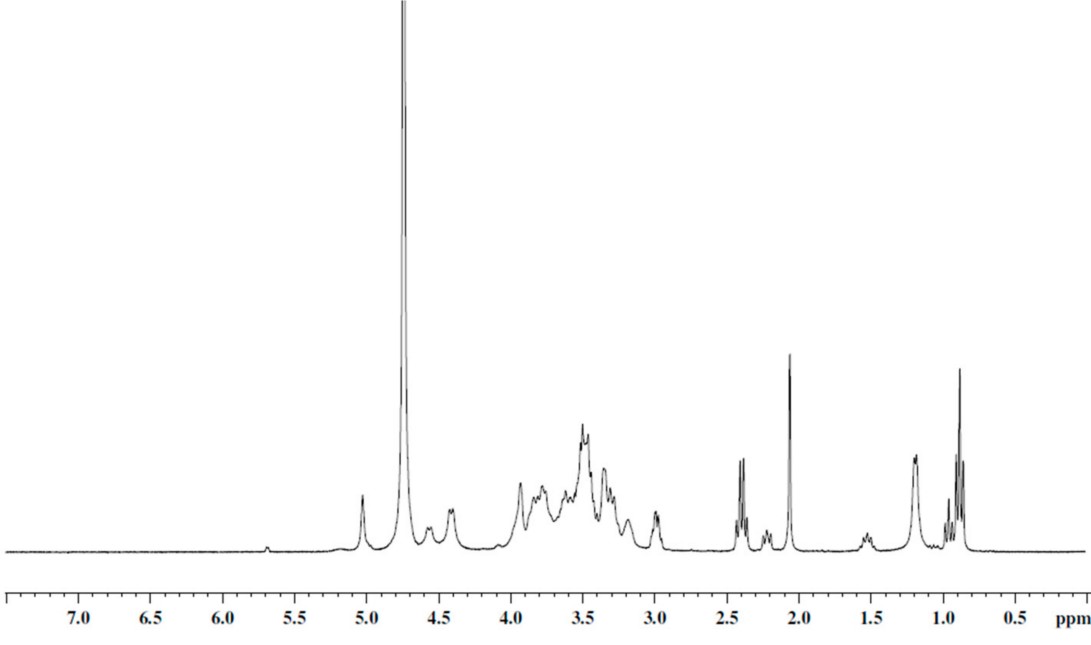

**Figure 1.** $^1$H NMR spectrum of GG-LA derivative, D$_2$O/NaOD, 300 MHz.

The sample was solubilized in deuterated water adding NaOD to allow a more deployed conformation of the polymer and facilitate proton resonance of the grafted lipoic acid functions. The ratio between the integrals of the peaks related to the 2H of lipoic acid ($\delta$ 2.97 (2H LA, -HO-CO-$CH_2$-$CH_2$-$CH_2$-$CH_2$-cCH-$CH_2$-$CH_2$-S-S-)) and the 3H of the GG ($\delta$ 1.2, $CH_3$ rhamnose) returned a degree of functionalization equal to a molar percentage of 20%.

The average molecular weight of the GG-LA copolymer was evaluated by means of SEC analysis in an aqueous mobile phase buffered at pH 9 (Table 1). The recorded values are in accordance with the functionalization of the sodium salt GG chains.

**Table 1.** Gellan gum (GG) sodium salt and GG-LA derivative weight average molecular weight, number average molecular weight, and polydispersity index.

| Sample | Mw (Da) | Mn (Da) | PDI |
|---|---|---|---|
| GG sodium salt | 43,600 | 31,500 | 1.38 |
| GG-LA | 70,600 | 54,412 | 1.30 |

### 3.2. Preparation and Characterization of Coated AuNRs

Although the cationic surfactant CTAB is a powerful stabilizing agent of lyophobic colloids, its toxicity avoids the use of CTAB-coated AuNRs in biomedical applications. For these reasons, gold nanoparticles need to be coated with molecules and/or macromolecules capable of physically stabilizing them as well as improving their cytocompatibility, allowing their use as drug delivery systems.

The first coating agent used in this study is lipoic acid (LA), a natural molecule whose structure is characterized by the presence of an intracycle disulfide bond and a carboxylic functional group. The preparation procedure of the AuNRs/LA systems involved the incubation of the freshly synthesized AuNRs with lipoic acid in the presence of a reducing environment for the addition of glucose. The thiol/disulfide redox pair was thus exploited to obtain a stable interaction with the gold surface.

According to data reported in the literature [34], gellan gum gels prepared in a basic environment have a reduced number of junction areas. In virtue of this, gel meshes are looser with increasing pH, an effect due to the greater electrostatic repulsive force exerted by the GG chains. Hence, the possibility of coating the nanorods in a basic environment has been investigated, with the aim to allow higher mobility of the polysaccharide chains and minimize aggregation phenomena. The preparation was therefore carried out at pH 7, to maintain an unfolded conformation of the GG during the coating procedure as well as to minimize the destabilization of the systems occurring at excessively high pH.

To calculate the amount of lipoic acid interacted with the gold nanorods surface, an AuNRs/LA preparation was carried out using LA labeled with the Texas Red Hydrazide (TRH) fluorescent probe. LA-TRH was synthesized via carbodiimide reaction between the carboxylic group of LA and the hydrazide group of TRH and used for the preparation of AuNRs/LA-TRH. The system produced was used to quantify LA, comparing the fluorescence intensity values recorded with a calibration curve obtained with TRH standards. The calculated lipoic acid content was 20% w/w with respect to the gold weight.

AuNRs/LA were thus coated with gellan gum, exploiting the abundance of hydroxyl groups to establish physical interactions with LA and the gold surface. The functionalization of AuNRs/LA with gellan gum was carried out incubating AuNRs/LA with an aqueous polymer dispersion at pH 7, leading to the formation of AuNRs/LA,GG. The rationale of gellan gum coating was the achievement of a second protective layer that further stabilizes the system, improving its cytocompatibility and allowing the future loading of bioactive molecules. This time, considering the heterogeneous sample, analysis of sulfur content of AuNRs/LA,GG was performed by X-ray photoelectron spectroscopy (XPS). The analysis showed a sulfur percentage of 0.97%, corresponding to 4.43% of LA with respect to the gold weight (Table 2).

**Table 2.** Atomic abundance percentages of gold nanorods (AuNRs)/LA,GG measured by X-ray photoelectron spectroscopy (XPS) analysis.

| Atom | % |
|------|-------|
| Au | 11.45 |
| S | 0.97 |
| C | 72.39 |
| O | 15.19 |

Finally, AuNRs/GG-LA were prepared by direct incubation of the nanorods pellet obtained after removal of the CTAB, with a polymeric dispersion of GG-LA in bidistilled water at pH 7 and in the presence of glucose.

Scanning electron microscopy (SEM) was carried out on CTAB-AuNRs and AuNRs/LA,GG to characterize Au nanorods in terms of morphology and size (Figure 2). SEM acquisitions confirmed the rod-like shape with an aspect ratio (length/width) of 3 and average dimensions of 54 nm × 18 nm (Figure 2a). Microscopies demonstrated that the coating was successfully performed (Figure 2b) and that no morphological changes occurred.

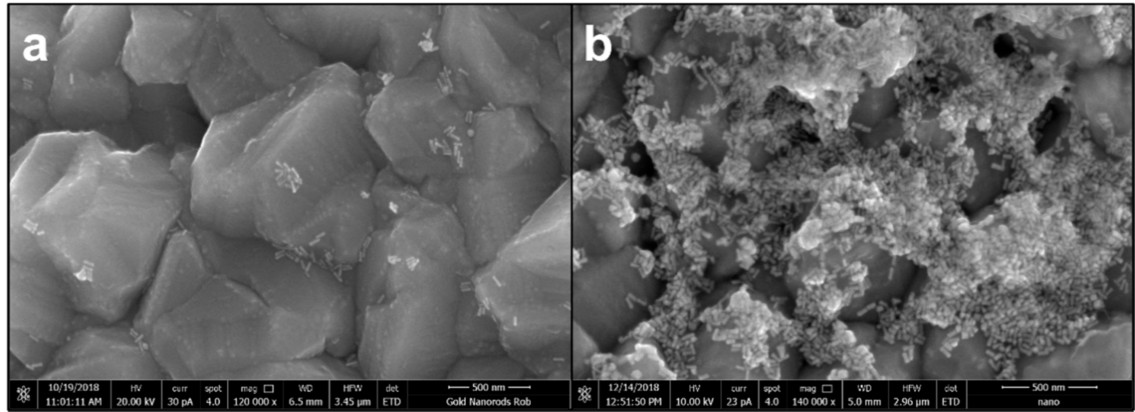

**Figure 2.** Scanning electron microscopy acquisitions of hexadecyltrimethylammonium bromide (CTAB)-gold nanorods (AuNRs) (**a**) and AuNRs/LA,GG (**b**) using magnifications 120,000× (**a**) and 140,000× (**b**). Scale bar: 500 nm.

UV–VIS–NIR characterization of the newly produced nanorods (Figure 3) show that all three types of coatings are able to stabilize nanorods dispersions, but with some differences. In particular, GG-LA seemed able to prevent aggregation considerably, as it is possible to appreciate from the narrow longitudinal plasmon ($\lambda_{max}$: 700 nm) of AuNRs/GG-LA. On the other hand, AuNRs/LA and AuNRs/LA,GG appeared properly stabilized by the coating agents used, although a slight widening of the longitudinal plasmons is registered. An interesting result lies in the differences between the maximum wavelengths of the three longitudinal plasmons. Indeed AuNRs/LA and AuNRs/LA,GG showed $\lambda_{max}$ at about 740 and 730 nm, respectively, suggesting the possibility to achieve a more efficient photothermal effect with respect to AuNRs/GG-LA after laser exposition (810 nm).

A study on the correct storage of the systems was carried out by performing lyophilization/redispersion tests in Milli-Q water and assessing their stability through UV–NIR analysis (Figure 4a).

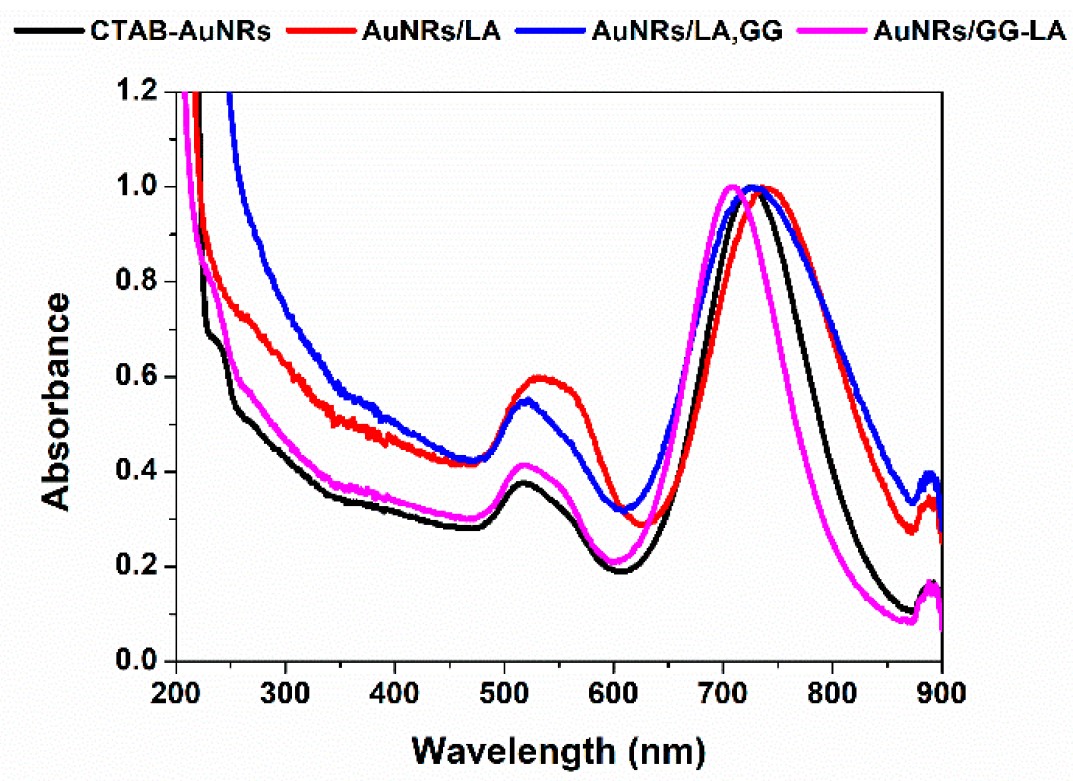

**Figure 3.** UV–VIS–NIR spectra of Milli-Q water dispersions of CTAB-AuNRs (black), AuNRs/LA (red), AuNRs/LA,GG (blue), and AuNRs/GG-LA (fuchsia).

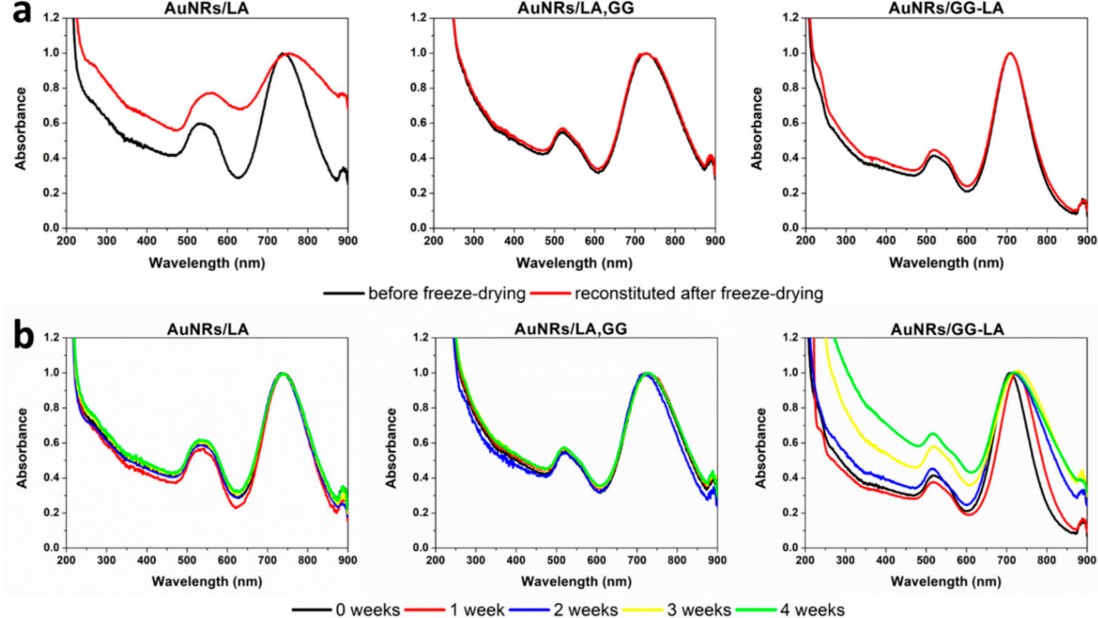

**Figure 4.** (**a**) VIS–NIR spectra before (black) and after (red) freeze-drying of AuNRs/lipoic acid (LA) (left), AuNRs/LA,GG (center), and AuNRs/GG-LA (right) in bidistilled water dispersion; (**b**) VIS–NIR spectra acquired every 7 days up to 1 month of AuNRs/LA (left), AuNRs/LA,GG (center), and AuNRs/GG-LA (right) in bidistilled water dispersion.

Freeze-dried AuNRs/LA systems showed evident destabilization after redispersion in aqueous medium, indicating the impossibility of preservation of the lyophilized product. Instead, it is possible

to note that both aqueous dispersions containing unmodified GG as well as derivatized GG, exhibit excellent stability after freeze-drying. These findings indicate that the presence of gellan gum can act as a cryoprotective agent of the proposed nanostructures. Consequently, it can be stated that AuNRs/LA,GG and AuNRs/GG-LA can be easily stored as lyophilizate powder, a feature that increases their handling and ensures easy administration following immediate redispersion.

The stability studies of the aqueous dispersion of AuNRs/LA, AuNRs/LA,GG, and AuNRs/GG-LA were carried out by UV analysis at 7 days intervals and for the total duration of 1 month.

Figure 4b illustrates that AuNRs/LA and AuNRs/LA,GG were stable throughout the analysis time, not registering significant variations in terms of shifting or widening of the longitudinal plasmonic peak. In contrast, the AuNRs/GG-LA nanorods show signs of destabilization since the end of the first week of incubation. The aggregation of the sample appears more pronounced during the rest of the analysis, signaling an enlargement of the plasmonic band contextual to the decrease in the absorbance ratio between the longitudinal (about 700 nm) and the transversal (about 516 nm) plasmonic resonance peak, confirming the progressive destabilization of the system. This finding denotes that is not recommended to store AuNRs/GG-LA in aqueous dispersion.

The nanorods produced were thus characterized in terms of hydrodynamic diameter and Zeta potential (Table 3).

**Table 3.** Dynamic light scattering measurements. Hydrodynamic diameters (numeric), polydispersity index (PDI) and ζ potentials of hexadecyltrimethylammonium bromide (CTAB)-AuNRs, AuNRs/LA, AuNRs/LA,GG, and AuNRs/GG-LA.

| Sample | Size (nm) | PDI | Zeta Potential (mV) |
| --- | --- | --- | --- |
| CTAB-AuNRs | 61.1 ± 26 | 0.55 | +41 ± 12.1 |
| AuNRs/LA | 97.5 ± 34 | 0.64 | +19.5 ± 6.76 |
| AuNRs/LA,GG | 146.7 ± 12 | 0.50 | −17.1 ± 4.96 |
| AuNRs/GG-LA | 173.0 ± 18 | 0.47 | −25.6 ± 5.77 |

Hydrodynamic diameters obtained are in agreement with the forecasted values. It should be noted that the CTAB-AuNRs, AuNRs/LA, AuNRs/LA,GG, and AuNRs/GG-LA series possess increasing dimensions, with hydrodynamic diameters enlarging as a function of the progressive coating. From the comparison between the AuNRs/LA,GG and AuNRs/GG-LA systems, it has been seen that GG-LA-coated systems possess a higher hydrodynamic diameter. This result is probably attributable to the slight tendency towards self-aggregation of the GG-LA copolymer, which can lead to an increase in the size of the system.

The Zeta potential data confirm the hypotheses formulated. In fact, the electrical surface potential decreases in accordance with the coating, registering strongly positive values for the system coated with the cationic surfactant and decreasing in modulus with the functionalization with LA. Finally, with the subsequent coating by gellan gum, a reversal of the sign of the Zeta potential was registered, reaching the maximum value for AuNRs/GG-LA. The data collected indicate that the starting nanorods have been successfully coated.

To evaluate the hyperthermia capabilities of gold nanorods after coating, the temperature variations of AuNRs/LA, AuNRs/LA,GG, and AuNRs/GG-LA dispersions were recorded during laser (810 nm) treatment. The study examined 3 different gold concentrations of nanosystems (10, 20, 35 μg·mL$^{-1}$) setting the laser power of $3.5 \times 10^{-3}$, $7 \times 10^{-3}$, and $14 \times 10^{-3}$ W/mm$^3$, in order to analyze the thermal behavior as a function of the irradiation time, the gold concentration, and the energy supplied (Figure 5).

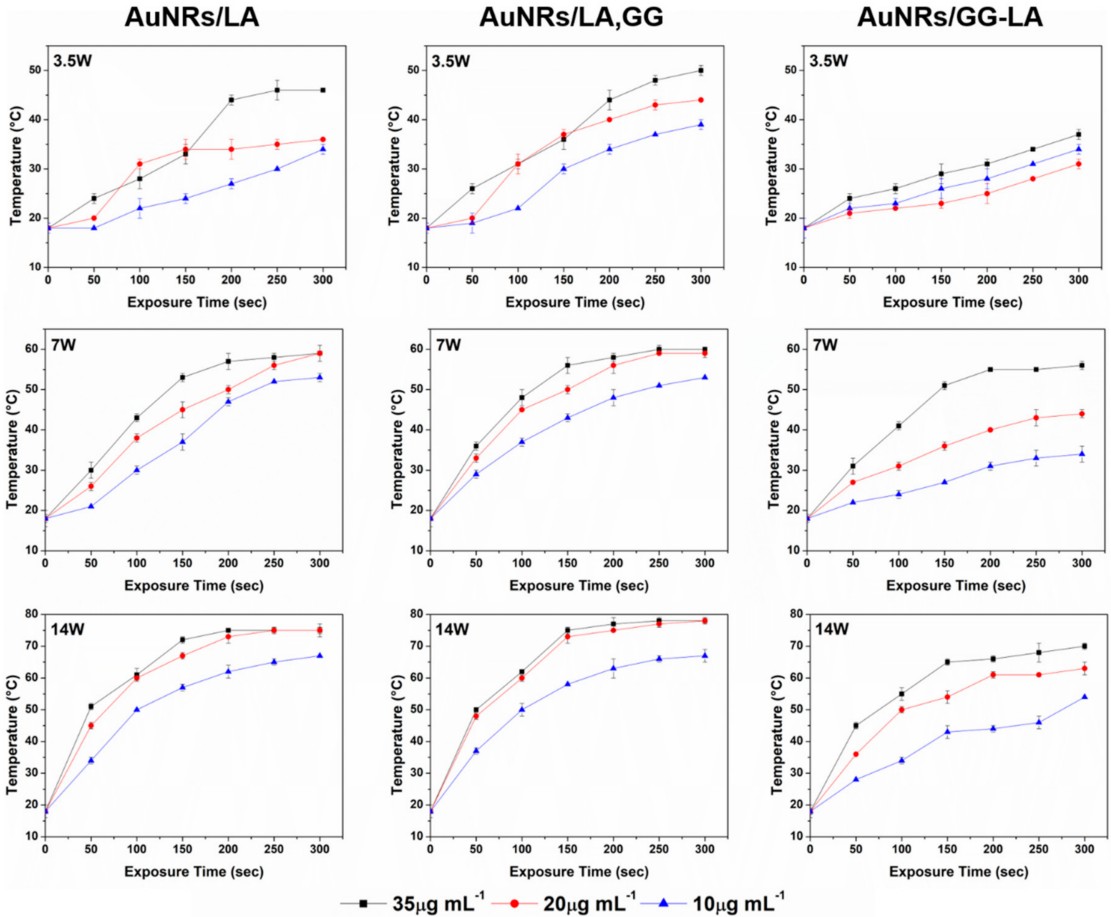

**Figure 5.** Time-dependent thermal rising of AuNRs/LA(left), AuNRs/LA,GG (center), and AuNRs/GG-LA (right) aqueous dispersions, at concentrations of 35, 20, and 10 $\mu g \cdot mL^{-1}$, during 300 s of 810 nm laser treatment with power of $3.5 \times 10^{-3}$ (first line), $7 \times 10^{-3}$ (second line), and $14 \times 10^{-3}$ W/mm$^3$ (third line).

The hyperthermia profiles of AuNRs/LA and AuNRs/LA,GG turned out to be quite similar under the same concentration and set power. AuNRs/GG-LA samples, although able to reach temperatures suitable for cancer PTT (powers $\geq 7 \times 10^{-3}$ W/mm$^3$), registered a slightly lower ability to increase the temperature of the surrounding medium. This finding could be explained by the presence of GG-LA, potentially capable of assuming a less deployed conformation compared to the GG sodium salt, acting presumably as a shield towards laser radiation. In this case, the energy absorbed by the systems would be reduced, consequently decreasing its transformation into thermal energy.

It can be noted that the application of the minimum laser power ($3.5 \times 10^{-3}$ W/mm$^3$) allowed to reach temperatures exploitable in photothermal therapy (45 and 50 °C, respectively) only for the highest concentrations of AuNRs/LA and AuNRs/LA,GG, in opposition to what occurs for AuNRs/GG-LA (maximum temperature reached: 37 °C).

Analyses conducted at $7 \times 10^{-3}$ W/mm$^3$ showed optimal results for AuNRs/LA and AuNRs/LA,GG regardless of the concentration tested, while AuNRs/GG-LA displayed good efficiency only at the highest concentration. Finally, the maximum power investigated ($14 \times 10^{-3}$ W/mm$^3$) returned exaggerated temperature values for AuNRs/LA and AuNRs/LA,GG, reaching 75 and 80 °C, respectively. Once again, AuNRs/GG-LA demonstrated a milder hyperthermia effect, reaching 70 °C at the maximum concentration analyzed and remaining in the range suitable for PTT at lower concentrations. Considering the data collected, the power of $7 \times 10^{-3}$ W/mm$^3$ appear the most suitable to obtain

effective treatment at the concentrations used, without incurring possible side effects due to the excessive induced thermal rise.

Taking into account the previous experiment, a new hyperthermia study was performed reproducing the setting of the future cellular experiment, but in absence of cell cultures. Analyses were conducted for a total duration of 100 s by fixing the power of the diode laser at $10 \times 10^{-3}$ W/mm$^3$ on samples at gold concentrations of 2.5 and 5 μg·mL$^{-1}$. These parameters were selected to maximize the hyperthermia efficiency of the selected gold concentrations and, concurrently, minimizing the possible undesired effects due to an excessive rising in temperature. In particular, the AuNRs/LA, AuNRs/LA,GG, and AuNRs/GG-LA samples were placed in 24-well plates and irradiated by laser, monitoring the temperatures of each well through an infrared camera.

Temperatures reported (Table 4) and images acquired by a thermal infrared camera (Figure 6) confirm the results of the previous study, but with some differences. AuNRs/LA,GG sample, once again, proves to be the most effective among the nanovehicle tested.

**Table 4.** Maximum temperatures recorded by thermal imaging camera during hyperthermia studies in 24-well plates (laser power of $10 \times 10^{-3}$ W/mm$^3$) of AuNRs/LA,GG, AuNRs/LA, and AuNRs/GG-LA samples corresponding to gold concentrations of 2.5 and 5 μg·mL$^{-1}$.

| Sample | Temperature (°C) versus Exposure Time (Seconds) | | | | | |
|---|---|---|---|---|---|---|
| | 0 s | 20 s | 40 s | 60 s | 80 s | 100 s |
| AuNRs/LA,GG 2.5 mg·mL$^{-1}$ Au | 25 °C | 33 °C | 36.9 °C | 40.3 °C | 43.4 °C | 46.4 °C |
| AuNRs/LA,GG 5 mg·mL$^{-1}$ Au | 25 °C | 37.7 °C | 42.6 °C | 47.1 °C | 51 °C | 54.5 °C |
| AuNRs/LA 2.5 mg·mL$^{-1}$ Au | 25 °C | 30.3 °C | 32.5 °C | 24.5 °C | 36.2 °C | 37.6 °C |
| AuNRs/LA 5 mg·mL$^{-1}$ Au | 25 °C | 34.5 °C | 38 °C | 41.2 °C | 43.9 °C | 45.6 °C |
| AuNRs/GG-LA 2.5 mg·mL$^{-1}$ Au | 25 °C | 29.9 °C | 31.6 °C | 32.9 °C | 34.4 °C | 35.6 °C |
| AuNRs/GG-LA 5 mg·mL$^{-1}$ Au | 25 °C | 31.5 °C | 34.3 °C | 37.1 °C | 39.3 °C | 41.5 °C |
| Blank (water) | 25 °C | 26 °C | 26 °C | 27.1 °C | 28.7 °C | 29.8 °C |

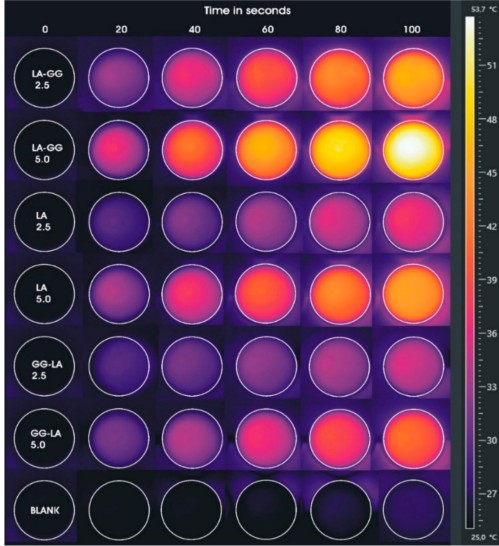

**Figure 6.** Images acquired by thermal imaging camera during laser hyperthermia studies in 24-well plates (laser power of $10 \times 10^{-3}$ W/mm$^3$) of AuNRs/LA,GG, AuNRs/LA, and AuNRs/GG-LA samples corresponding to gold concentrations of 2.5 and 5 μg·mL$^{-1}$.

The comparison between the maximum temperatures reached by AuNRs/LA,GG and AuNRs/LA shows that the former system is far more efficient than the latter (reaching temperatures of 54.5 and 45.6 °C, respectively). The greater difference recorded in this study compared to the previous one is attributable, in first instance, to the temperature detection method. In the experiment carried out by setting three different powers of the laser treatment, the measurements by optical fiber allowed to obtain the average temperature value in each well. On the contrary, temperatures, acquired through infrared camera and shown in Table 4, represent the maximum temperatures detected. A further explanation of the differences found might be the use of lower concentrations. In fact, in view of the considerable dependence between gold concentration and thermal rise, it is possible that the use of low sample concentrations may allow more detailed observation of the gap between the two hyperthermic behaviors. However, it is not surprising that the AuNRs/GG-LA sample shows the mildest hyperthermic effect, touching the maximum temperature of 41.5 °C.

The collected data were used as reference temperatures for the hyperthermia experiment later conducted on HCT116 tumor cell line.

### 3.3. In Vitro Studies

The cytocompatibility of the coated nanorods produced was tested on HCT116 human colon cancer cell line, via MTS assay. In particular, the metabolic activity of the cells was measured after 24 or 48 h incubation with the three types of coated AuNRs (Figure 7).

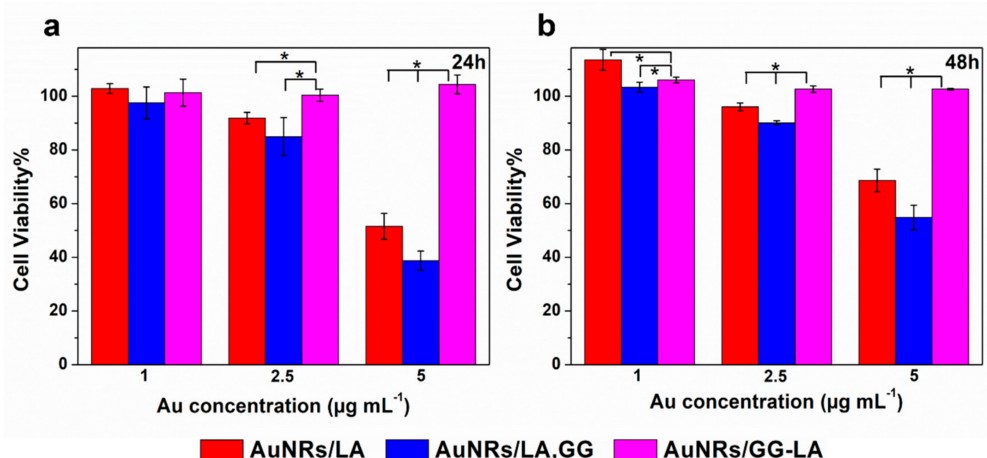

**Figure 7.** Cell viability of human colorectal cancer cell line (HCT116) line after 24 h (**a**) or 48 h (**b**) incubation with AuNRs/LA (red), AuNRs/LA,GG (blue), and AuNRs/GG-LA (fuchsia) at gold concentrations of 1, 2.5, and 5 µg·mL$^{-1}$. Statistical significance (Student's two-tailed *t* test): * $p < 0.05$.

All samples at the lowest concentrations tested appear cytocompatible during the whole experiment. The highest concentration of AuNRs/LA and AuNRs/LA,GG (Au: 5 µg·mL$^{-1}$) show medium/moderate toxicity, recording the minimum cell viability value after 24 h incubation with AuNRs/LA,GG (about 40%). On the other hand, no significant variations in cell viability are appreciated after AuNRs/GG-LA incubation, constantly recording high cytocompatibility values in all conditions tested. It could be hypothesized that the derivative that coats AuNRs/GG-LA is able to form a compact shell on the nanoparticles, which might lead to a reduction in the interactions with the cells and, as a consequence, to cell viability values almost overlapping the control samples. The comparison between AuNRs/LA and AuNRs/LA,GG, showed comparable values throughout the analysis. An interesting result was, however, highlighted after 48 h of incubation, as the incipient toxicity registered during the first 24 h of incubation seems to decrease in the following 24 h. In fact, cell viability data collected after 48 h appear a little increased in comparison to the first time point. The results obtained could suggest that the incubation with nanorods might disturb cell milieu, causing a decrease in metabolic activity.

However, this settling phase is presumably transient, since during the second day, the cultured cells seem to enter in a recovery phase of the metabolic activity. Data collected are encouraging since they suggest a good overall cytocompatibility of the system tested.

The in vitro hyperthermia study involved the incubation of HCT116 human colon cancer cells with the three types of systems at a concentration of 5 μg·mL$^{-1}$. Cell viability after laser treatment (810 nm, $10 \times 10^{-3}$ W/mm$^3$, and 100 s), was therefore evaluated at 20 s intervals using MTS colorimetric assay and expressed as a percentage with respect to the untreated cells used as control.

As reported in Figure 8, the comparison between the three types of nanostructures revealed cytotoxicity in accordance with the hyperthermia studies.

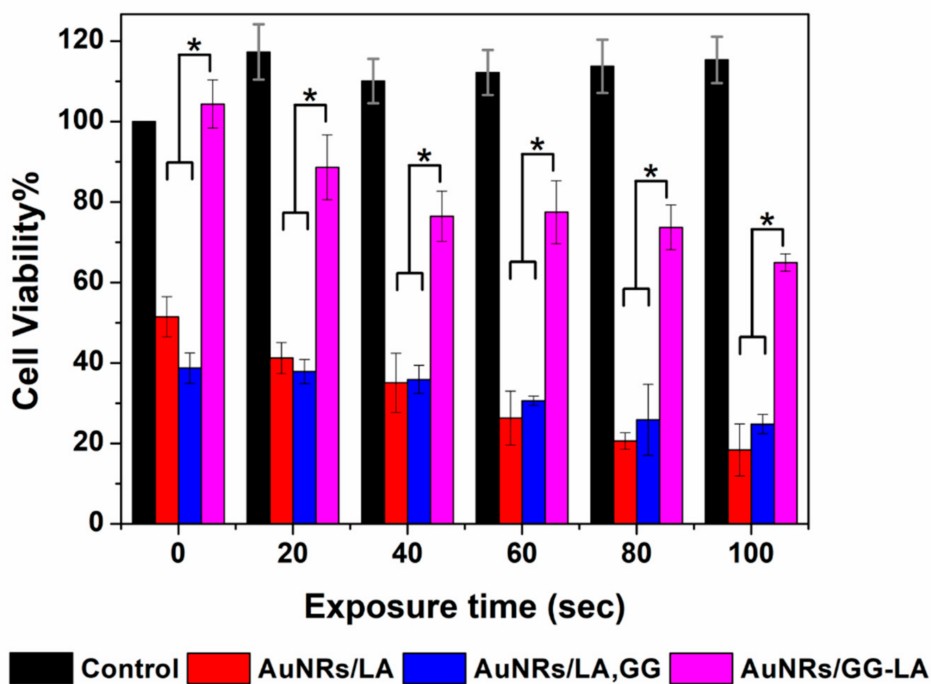

**Figure 8.** Cell viability of HCT116 line incubated with DMEM (black), AuNRs/LA (red), AuNRs/LA, gellan gum (GG) (blue) (Au concentration: 5 μg·mL$^{-1}$) and AuNRs/GG-LA (fuchsia) after laser treatment (810 nm, $10 \times 10^{-3}$ W/mm$^3$, and 100 s), versus irradiation time. Statistical significance (Student's two-tailed *t* test): * $p < 0.05$.

To confirm the ineffectiveness of the sole laser treatment, the viability data of cells irradiated without nanorods incubation have been reported. It is evident that the laser treatment is not harmful at all, detecting a high vitality during the whole experiment.

Once again, AuNRs/LA and AuNRs/LA,GG demonstrate to induce a significant decrease in vitality. On the contrary, the data of metabolic activity after AuNRs/GG-LA treatment appear definitely high, for all the time points analyzed. Even this time, it is conceivable that the GG-LA coating is able to shield the energy administered, suggesting that AuNRs/GG-LA are not ideal for photothermal therapy.

The AuNRs/LA and AuNRs/LA,GG nanorods have shown good efficacy in reducing cell viability. Cell death induced by hyperthermia is consistent, recording eradication percentages of 82% and 75%, respectively, for AuNRs/LA and AuNRs/LA,GG.

Finally, the residual cell viability after laser irradiation was confirmed by LIVE/DEAD assay (Figure 9). After treatment lasting 100 s, the contents of the wells were incubated with the fluorescent probes calcein AM and ethidium homodimer-1, capable of discriminating respectively the live cells (green) from those that present irreparable damage to the cell membrane (red). The images were, therefore, acquired using a fluorescence microscope using a 5× magnification.

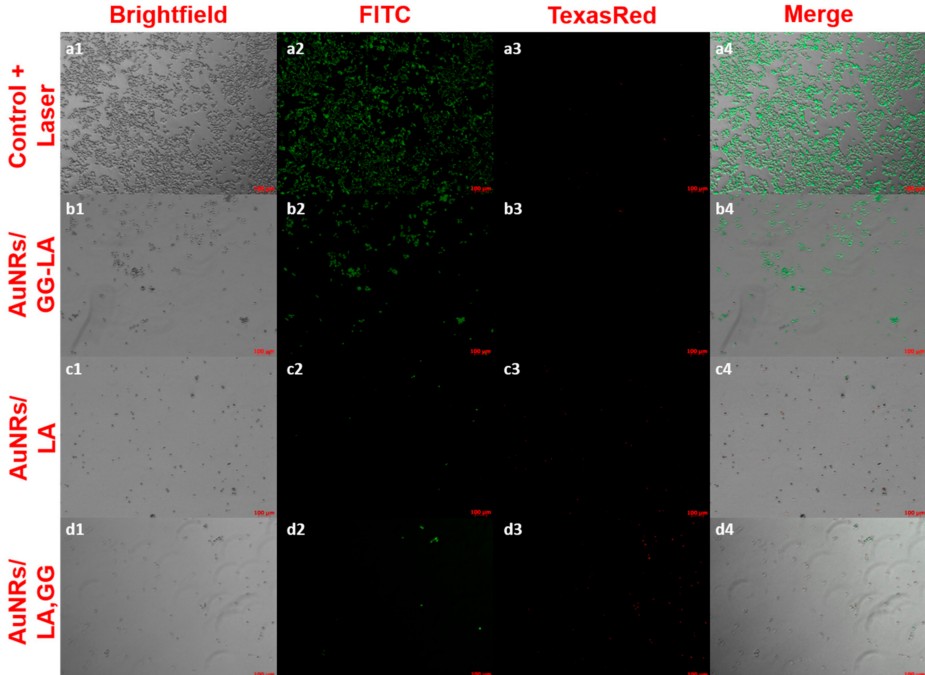

**Figure 9.** LIVE/DEAD assay fluorescence microscopy images on laser-treated HCT116 cell culture (810 nm, $10 \times 10^{-3}$ W/mm$^3$, and 100 s). Here, 5× magnifications of: control + laser (**a**), AuNRs/GG-LA (**b**), AuNRs/LA (**c**), and AuNRs/LA,GG (**d**) are used. Acquisition channels: brightfield (1), fluorescein isothiocyanate (FITC) (2), Texas Red (3), and merge (4); scale bar: 100 μm.

The images in Figure 9 are in full agreement with the results of the MTS test, underlining the effectiveness of the AuNRs/LA and AuNRs/LA,GG systems. The results obtained appear decidedly promising, indicating a possible use of these two nanoconstructs for photothermal therapy in vivo.

## 4. Conclusions

Multimodal devices able to combine therapy and diagnostic are crucial to achieve efficient patient-tailored therapy. Although gold nanorods are potentially capable of fulfilling this aim, current research is still focusing on new biocompatible materials capable to allow proper in vivo use and to confer additional beneficial attributes. The query of this work was to test the natural materials lipoic acid and gellan gum as coating agents of gold nanorods, investigating how different coating strategies could influence the product properties and suitability in PTT. Briefly summarizing data collected, AuNRs were successfully coated with LA and GG leading to the formation of the anisotropic nanosystems AuNRs/LA, AuNRs/LA,GG, and AuNRs/GG-LA. AuNRs coated with the ester derivative of GG and LA (AuNRs/GG-LA) denoted some tendency to aggregate, displaying reduced stability in aqueous medium. More importantly, AuNRs/GG-LA, despite capable to reach temperatures suitable for hyperthermia, shown a far lower efficacy in in vitro PTT studies. On the other hand, AuNRs/LA and AuNRs/LA,GG dispersions demonstrated optimal stability and prominent photothermal properties that can suggest a use for in vivo PTT of cancer. Nevertheless, the presence of GG allows the possibility to store AuNRs/LA,GG as lyophilized powder and, in addition, GG shell could be employed to load bioactive molecules for drug delivery purposes. For these reasons, AuNRs/LA,GG represent the most promising choice for further use in in vivo photothermal therapy of cancer.

**Author Contributions:** All the authors contributed individually to the study and manuscript production with the following roles: M.L., conceptualization, data curation, supervision and writing; P.V.: preparation, characterization of nanosystems and writing; L.T.: thermal imaging analysis; G.C.: manuscript revision. All authors have read and agreed to the published version of the manuscript.

**Funding:** This research received no external funding.

**Acknowledgments:** Authors thank the ATeN Center of University of Palermo—Laboratory of Preparazione e Analisi di Biomateriali, for SEM and XPS analysis of nanorods.

**Conflicts of Interest:** The authors declare no conflict of interest.

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
