# Peer review of "Preparation and Characterization of Gold Nanorods Coated with Gellan Gum and Lipoic Acid"

_applsci, doi:10.3390/app10238322_

Round 1

Reviewer 1 Report

  Varvara et al., have presented a very interesting paper employing coated gold nanorods for biomedical application through the use of photothermal therapy. The formed gold nanorods and the lipoic acid and gellan gum are well characterised and their ability to perform photothermal therapy on human colon cancer cells in commendable.    However, in my opinion there are some measurements and controls that would greatly enhance the impact of this paper, which I will outline below. Secondly, there are some typographical changes that must be made before publication of the manuscript.    Major Changes:  1/ The coated gold nanorods are well characterised. However, there does not seem to be a comparative optical characterisation of the original CTAB stabilised nanorods. Is it possible to have a single graph which compares the optical properties of each system as a function of surface functionality (comparison of LSPR peak positions, relative intensities)?    2/ In comparison to the coated particles, how do the uncoated gold nanorods perform in the freeze-drying/reconstitution experiments? Does the surface functionality add or remove redispersibility?   3/ Is it possible to perform electron microscopy on the coated gold nanorods? It would be interesting to see if there is any morphological changes post treatment.    4/ The zeta potential measurements reveal a residual positive charge on the lipoic acid functionalised gold nanorods. Could this possibly be indicative of retention of CTAB molecules on the surface? Would it be possible to repeat the lipoic acid functionalisation 2 or 3 times in order to see if this charge can be driven to an overall negative value, as would be expected with lipoic acid stabilised gold nanoparticles?   5/ The cell viability assays show considerable toxicity for two of the gold nanorod families, in particular the LA functionalised. Given the zeta potential results, the authors have not addressed whether this could be due to the known toxicity of CTAB, which may still be present on the surface of the gold nanorods. This toxicity does not seem to be accounted for in the hyperthermia studies, where the gold nanorod families which perform the best also happen to be the ones with the largest toxicity towards the cell line.    6/ Is the figure caption for Figure 7 correct? Is the fuschia bar the DMEM control?   Minor Corrections: 1/ The introduction could use some clarification/change in the aims section, which does not read smoothly.    2/ Throughout the manuscript the grammar and syntax of the language could be improved to make it read more clearly.     3/ Throughout the manuscript there are incidents where there is no space between the number and the unit when describing values. Similarly, capital letters are used in units when they should not be (e.g. KDA should be kDa and MΩ should be mΩ    

Author Response

Reply to reviewer 1.

Here enclosed you'll find a point-by-point details of the revisions in the manuscript and author's responses to the reviewer's comments:

1/ The coated gold nanorods are well characterised. However, there does not seem to be a comparative optical characterisation of the original CTAB stabilised nanorods. Is it possible to have a single graph which compares the optical properties of each system as a function of surface functionality (comparison of LSPR peak positions, relative intensities)? 

1/ According with reviewer suggestion, authors think that a comparison between UV spectra of the starting CTAB-AuNRs and the proposed nanostructures in aqueous dispersion could better clarify their optical properties. The new graph (new Figure 3) and the related discussion (lines 47-56) were thus added in the text accordingly.

2/ In comparison to the coated particles, how do the uncoated gold nanorods perform in the freeze-drying/reconstitution experiments? Does the surface functionality add or remove redispersibility?

2/ The complete removal of the starting stabilizing agent (CTAB), without adding any other coating material, causes a rapid aggregation of gold nanorods after the centrifugation step. This destabilization is evident (visible to the naked eye) and it does not allow redispersion, even before the freeze-drying procedure. On the other hand, authors think that comparison on re-dispersibility behavior between the proposed systems and CTAB stabilized AuNRs could be not appropriate since CTAB coated AuNRs are not exploitable in biomedical applications without any modification that implies the removal of the majority of CTAB. For these reasons, re-dispersibility tests after freeze-drying were not performed on naked AuNRs or CTAB-AuNRs.

3/ Is it possible to perform electron microscopy on the coated gold nanorods? It would be interesting to see if there is any morphological changes post treatment.

3/ Thank you for the comment. Authors agree with the reviewer that the addition of SEM acquisitions on coated nanorods could improve the manuscript. SEM panel (Figure 2) was modified adding coated systems. Accordingly, a text was addeded at lines 338-342 of revised manuscript.

4/ The zeta potential measurements reveal a residual positive charge on the lipoic acid functionalised gold nanorods. Could this possibly be indicative of retention of CTAB molecules on the surface? Would it be possible to repeat the lipoic acid functionalisation 2 or 3 times in order to see if this charge can be driven to an overall negative value, as would be expected with lipoic acid stabilised gold nanoparticles? 

4/ This is of course a good observation. The preparation of AuNRs/LA was carried out three times without registering any difference in the zeta potential values. We agree with the reviewer in saying that a residual of CTAB may be still present on AuNRs/LA samples and it is responsible for the positive value of the Zeta Potential. Indeed, it is reported in literature that CTAB residuals on AuNRs surface are still present even after the coating procedure [García, I. et al (2015). Residual CTAB ligands as mass spectrometry labels to monitor cellular uptake of Au nanorods. The Journal of Physical Chemistry Letters, 6(11), 2003-2008.;Kinnear, C. et al.(2013). Gold Nanorods: Controlling Their Surface Chemistry and Complete Detoxification by a Two-Step Place Exchange. Angew. Chem., Int. Ed., 52, 1934−1938.]

In this regard, authors think that lipoic acid coating is able to partially neutralize the positive charge imparted by CTAB residuals, although not completely, reducing the absolute value of the Zeta potential.

5/ The cell viability assays show considerable toxicity for two of the gold nanorod families, in particular the LA functionalised. Given the zeta potential results, the authors have not addressed whether this could be due to the known toxicity of CTAB, which may still be present on the surface of the gold nanorods. This toxicity does not seem to be accounted for in the hyperthermia studies, where the gold nanorod families which perform the best also happen to be the ones with the largest toxicity towards the cell line. 

5/ Authors thank reviewer for the comment. Cytotoxicity without laser treatment illustrated in Figure 7 (Figure 6 before revision) shows a cytotoxic effect of AuNRs/LA and AuNRs/LA,GG at the highest concentration used, especially after 24h of incubation. We agree with the reviewer thinking that CTAB residues might be cytotoxic. Nevertheless, the results obtained show that not only AuNRs/LA, but also AuNRs/LA,GG reduce the cell viability, although AuNRs/LA,GG Zeta potential was found to be clearly negative (-17.1 mV). Hence, it is the authors opinion that the cytotoxicity registered is not a primary consequence of the positive value of Zeta potential.

The hyperthermia studies on cells explore the ability of the coated nanorods to reduce cell viability upon laser exposition. In Figure 8 (Figure 7 before revision) the first set of data (0 on x-axis) reports cell viability values after 24h of incubation whithout laser irradiation. These values may be due to possible CTAB cytotoxic effect. Actually, in all the other set of data in figue 8, it is evidenced the improvement of cytotoxicity related to photothermal effect.

6/ Is the figure caption for Figure 7 correct? Is the fuschia bar the DMEM control? 

6/ Authors thank the reviewer for this point. Caption of Figure 7 was modified in order to make it correct and in agreement with the Figure 7 legend.

Minor Corrections:

1/ The introduction could use some clarification/change in the aims section, which does not read smoothly. 

1/ The description of the aim of the work was edited to make it clearer to readers (lines 87-90 in the revised version).

2/ Throughout the manuscript the grammar and syntax of the language could be improved to make it read more clearly.

2/ Thank you for the comment. The manuscript was once again spell-checked.

3/ Throughout the manuscript there are incidents where there is no space between the number and the unit when describing values. Similarly, capital letters are used in units when they should not be (e.g. KDA should be kDa and MΩ should be mΩ).

3/ Following the reviewer’s comment, the authors removed typographical error throughout the text.

Reviewer 2 Report

In this contribution Varvara and coworkers present the synthesis of different types of gold nanorods stabilized with different reagents. The Authors performed detailed characterization of the obtained Au nanorods using scanning electron microscopy, X-ray photoelectron spectroscopy, VIS-NIR spectroscopy, and dynamic  light  scattering. Additionally, the evaluation of the cytotoxic effect of the synthesized Au nanorods on human colon cancer  cell line using MTS assay were conducted. In my opinion all described experiments have been performed properly and the conclusions are correct.

However, there are some points which could be developed:

  1. In my opinion the discussion based on the XPS analysis is a little bit poor. Authors should provide the XPS spectra together with detailed interpretation which could give information about the stabilized layer forming on the synthesized nanorods.
  2. The MTS assay results show very interesting dependencies. However, the more deeply explanation of them should be provided. The AuNRs/GG-LA indicate the low cytotoxic effect in comparison with the AuNRs/LA and AuNRs/LA,GG. I am wondering if the Authors are sure that the AuNRs/GG-LA penetrate the cells. Maybe they did not get efficiently inside the cells and this is the reason why the cytotoxic effect is poorly visible.
  3. The applied color schemes in Figures 6 and 7 should be consistent. Author should also avoid black color because error bars are not sufficiently visible.

Author Response

Reply to reviewer 2.

Here enclosed you'll find a point-by-point details of the revisions in the manuscript and author's responses to the reviewer's comments:

  • In my opinion the discussion based on the XPS analysis is a little bit poor. Authors should provide the XPS spectra together with detailed interpretation which could give information about the stabilized layer forming on the synthesized nanorods.
  • Authors thank the reviewer fot the suggestion. Nevetheless, it is the authors’ opinion, that the addition of the XPS spectra and their detailed interpretation, is distant from authors purpose. XPS analysis was performed with the precise aim to calculate the lipoic acid weight percentage on coated gold nanorods. In the text, the authors also reported a second analysis (fluorescence quantification) aimed to quantify the percentage of lipoic acid on the coated nanorods. Thus, XPS analysis was not carried out with the intention to provide further information about the nanorods stabilizing layer.

  • The MTS assay results show very interesting dependencies. However, the more deeply explanation of them should be provided. The AuNRs/GG-LA indicate the low cytotoxic effect in comparison with the AuNRs/LA and AuNRs/LA,GG. I am wondering if the Authors are sure that the AuNRs/GG-LA penetrate the cells. Maybe they did not get efficiently inside the cells and this is the reason why the cytotoxic effect is poorly visible.
  • The authors fully agree with the reviewer. A possible explanation for the low cytotoxicity registered after AuNRs/GG-LA incubation with cells might be a lack of interaction between cells and nanorods. In this regard, the discussion about this point can be found in the Results and Discussion section (lines 479-483). In this work, the authors did not performed uptake studies evaluating the ability of the proposed nanosystems to penetrate the cells, therefore we have not the proof that AuNRs/GG-LA get efficiently or not inside the cells. The main applicative aim is, indeed, to examine coated nanorods' behavior upon laser irradiation in the presence of cells. This technique does not necessarily require cell penetration, whilst it is sufficient the presence of the photothermal agent (in this case nanorods) in the proximity of the cells, allowing the rise in temperature of the cell milieu.

  • The applied color schemes in Figures 6 and 7 should be consistent. Author should also avoid black color because error bars are not sufficiently visible.
  • Thank you for the comment. The authors think that these changes could improve the clarity of the manuscript. Colors in new Figure 7 and 8 (in the revised manuscript) were modified to make them consistent. Furthermore, error bars on black series in new Figure 8 were edited in order to make them more visible.

Reviewer 3 Report

In this manuscript, the authors have prepared 3 gold nanorods using lipoic acid
 (AuNRs/LA), a layer by layer coating of LA and polysaccharide chains of GG (AuNRs/LA,GG) and  finally, the covalent derivative obtained from esterification between GG and LA (AuNRs/GG-LA) in comparison of the CTAB-AuNRs, whose ctytotoxicity has been proved. The authors not only considered the cytotoxicyt, but also evaluated the PTT application. However, some suggestions are as following:

  1. The authors have evaluated the cytocompatibility of these AuNRs. However, the authors only assessed the cell viability of cancer cells HCT116. The normal cells are more appropriate for the compatibility tests.
  2. Line 33-35: please check the sentence.

Author Response

Reply to reviewer 3.

Here enclosed you'll find a point-by-point details of the revisions in the manuscript and author's responses to the reviewer's comments:

The authors have evaluated the cytocompatibility of these AuNRs. However, the authors only assessed the cell viability of cancer cells HCT116. The normal cells are more appropriate for the compatibility tests.

Authors agree with the reviewer comment that it is praxis to also check systems cytocompatibility on normal cells. Nevertheless, in this study, the main aim was to demonstrate the potential use of the proposed coated nanorods as tools in cancer therapy. For this reason, in vitro studies were performed only on the colorectal cancer cell line HCT116, evaluating the cell viability before (Figure 7 of revised manuscript) and after (Figure 8 of revised manuscript) laser treatment. In this way, it was possible to compare the effect of laser irratiation on cancer cell death. Accordigly with reviewer, it is authors intention to perform future in vitro and in vivo biological evaluations to appreciate the selective effect of the proposed coated systems in photothermal therapy of cancer.

Line 33-35: please check the sentence.

Thank you for the suggestion. Lines 33-35 were edited to make the sentence clearer.

Round 2

Reviewer 1 Report

The authors have addressed issues raised in the review process sufficiently and I feel the manuscript is suitable for publication.